# 4-Methylumbelliferone Modulates CAIX to Mitigate Hypoxia-Driven Dysregulation and Enhance PD-1 Immunotherapy in Lung Cancer

**DOI:** 10.3390/ijms262110427

**Published:** 2025-10-27

**Authors:** Mariel Fusco, Carlos Rafael Picón, Marco Aurelio Diaz, Juan Bayo, Paula Constanza Arriola Benitez, Flavia Piccioni, Noelia Gómez, Mara Stinco, Javier Martínez Martinez, José Nicolás Minatta, Ricardo Amorín, Martina Villar, Valentina Sole, Ignacio Cassol, Mauricio De Marzi, Manglio Miguel Rizzo, María Florencia Mercogliano, Mariana Malvicini

**Affiliations:** 1Cancer Immunobiology Laboratory, Facultad de Ciencias Biomédicas, Instituto de Investigaciones en Medicina Traslacional (IIMT), CONICET-Universidad Austral, Av. Presidente Perón 1500 Derqui-Pilar, Buenos Aires B1629ODT, Argentina; mfusco-conicet@austral.edu.ar (M.F.); diazmarcoaurelio@gmail.com (M.A.D.); carriola-conicet@austral.edu.ar (P.C.A.B.); fpiccioni@austral.edu.ar (F.P.); nugomez9@gmail.com (N.G.); marastinco@gmail.com (M.S.); mrizzo@cas.austral.edu.ar (M.M.R.); 2Department of Clinical Oncology, Hospital Universitario Austral, Av. Presidente Perón 1500 Pilar, Buenos Aires B1629AJH, Argentina; cpicon@cas.austral.edu.ar (C.R.P.); javimart31@gmail.com (J.M.M.); jminatta@cas.austral.edu.ar (J.N.M.); ramorin@cas.austral.edu.ar (R.A.); 3Programa de Hepatología Experimental y Terapia Génica, Facultad de Ciencias Biomédicas, Instituto de Investigaciones en Medicina Traslacional (IIMT), CONICET-Universidad Austral, Av. Presidente Perón 1500 Pilar, Buenos Aires B1629AJH, Argentina; jbayofina@austral.edu.ar; 4Facultad de Ingeniería, LIDTUA-CIC, Universidad Austral, Mariano Acosta 1611 Pilar, Buenos Aires B1629WWA, Argentina; martina.villar@ing.austral.edu.ar (M.V.); valentina.sole@ing.austral.edu.ar (V.S.); icassol@austral.edu.ar (I.C.); 5Grupo de Investigaciones Básicas y Aplicadas en Inmunología y Bioactivos (GIBAIB), Instituto de Ecología y Desarrollo Sustentable (INEDES), CONICET-Universidad Nacional de Luján, Luján B6700, Argentina; mauriciocesardemarzi@gmail.com; 6Departamento de Ciencias Básicas, Universidad Nacional de Luján, Luján B6700, Argentina; 7FOCIS Center of Excellence—Vaccines and Immuno Therapies against Infections and Cancers (VITIC), Menomonee Falls, WI 53051, USA; 8Laboratorio de Inmunología Tumoral, Instituto de Biología y Medicina Experimental (IBYME-CONICET) Vuelta de Obligado 2490, Buenos Aires C1428ADN, Argentina

**Keywords:** hypoxia, carbonic anhydrase IX (CAIX), 4-methylumbelliferone (4Mu), tumor microenvironment

## Abstract

Hypoxia is a hallmark of solid tumors, driving metabolic reprogramming and immune evasion. In lung cancer, hypoxia-induced activation of carbonic anhydrase IX (CAIX) promotes lactate accumulation and extracellular acidification, fostering an immunosuppressive tumor microenvironment (TME). Analysis of public datasets revealed that patients with high CAIX expression exhibited significantly reduced median survival (*p* < 0.001). Moreover, CAIX correlated with HIF-1α, PD-L1, and immunosuppressant molecules, linking hypoxia-driven metabolic alterations with immune dysfunction. Here, we evaluated the capacity of 4-methylumbelliferone (4Mu) to counteract these effects and enhance antitumor immunity. In vitro, hypoxia increased CAIX and monocarboxylate transporter -4 (MCT4) expression in lung carcinoma cells, elevated lactate release, and reduced extracellular pH while promoting an M2-like macrophage profile and impairing antigen-specific splenocyte proliferation (*p* < 0.01). Treatment with 4Mu downregulated CAIX expression, restored extracellular pH, decreased lactate secretion, and rescued lymphocyte proliferation (*p* < 0.01). In vivo, 4Mu reduced CAIX expression, shifted macrophage polarization toward a pro-inflammatory phenotype, and enhanced CD8+ T cell infiltration. 4Mu was safe and well tolerated, and notably, combined with anti-PD-1 therapy, it synergistically inhibited tumor growth and increased both CD4+ and CD8+ T cell infiltration. These findings support 4Mu as a metabolic modulator capable of mitigating CAIX-driven acidosis and improving the efficacy of immunotherapy in lung cancer.

## 1. Introduction

Lung cancer remains the leading cause of cancer-related mortality worldwide [1]. Immunotherapy using the immune checkpoint inhibitors (ICIs) anti-PD-1 (Programmed Death 1), anti-PD-L1 (Programmed Death Ligand 1), and anti-CTLA4 (Cytotoxic T-lymphocyte associated protein 4) is currently the standard of treatment for patients with advanced stages (IIIb and IV) of non-small cell lung cancer (NSCLC) [2]. These agents have been shown to increase overall survival (OS) and progression-free survival (PFS) compared with conventional chemotherapy in patients without driver mutations [3]. However, clinical outcomes remain poor, with response rates of about 20–30% [4].

A possible mechanism of resistance to immunotherapy is the low oxygen concentration or hypoxia, a common feature in the tumor microenvironment (TME) of solid tumors. Hypoxia profoundly influences cancer cell metabolism, immune evasion, and resistance to therapy [5]. Under low oxygen availability, hypoxia-inducible factor 1-alpha (HIF1α) orchestrates the transcription of genes involved in glycolysis, acid–base regulation, and angiogenesis [6]. One of the hypoxia-related metabolic changes is the use of other pathways to metabolize glucose, the conversion of pyruvate to lactate by Lactate Dehydrogenase (LDH), and the resulting intracellular acidification induced by changes in the lactate concentration ranges [7]. Hypoxic tumor cells regulate the internal pH through the modulation of cell surface molecules such as extracellular carbonic anhydrases (CAIX and CAXII) and monocarboxylate transporters (especially MCT4), as proton export and lactate extrusion mechanisms responsible for extracellular acidification, which drives malignant progression [8,9], increases cell survival and invasiveness, and contributes to modifying immune cell function [10]. Furthermore, hypoxia in the TME induces the infiltration of immune-suppressing cells, such as regulatory T cells (Tregs), myeloid-derived suppressor cells, and tumor-associated macrophages (TAMs), as well as the expression of PD-L1 on tumor cells, suppressing T cell activity [11]. Successful immunotherapy requires the inhibition of immunosuppressive mechanisms generated by tumors and the efficient stimulation of specific T cells. The repertoire of reprogramming-induced effects that sustain tumor survival and invasion has not yet been fully explored as mechanisms of resistance to immunotherapy or as therapeutic targets. Additionally, they may emerge as potential tools for the prognostic and predictive assessment of patients with advanced disease; more importantly, targeting hypoxia-induced metabolic pathways could represent a promising strategy to enhance the efficacy of ICIs. Despite the recognized role of hypoxia-driven metabolic reprogramming in shaping the immunosuppressive TME, the specific contribution to resistance to PD-1 blockade remains to be elucidated. 4-Methylumbelliferone (4Mu), a coumarin derivative, has been shown to modulate CAIX activity and immune populations in the TME [12,13]. Our aim was to investigate whether 4Mu could reverse metabolic and immune alterations and improve the response to PD-1 inhibition in lung cancer.

In this study, we investigated the relationship between hypoxia-associated metabolic reprogramming and immune regulation in NSCLC patients from public databases. We also evaluated the impact of hypoxia on human and murine lung cancer cells, the effect of cancer cell-conditioned medium on macrophage polarization, and lymphocyte proliferation in vitro. Moreover, we showed that 4Mu reduced lactate concentration and restored pH, modified CAIX expression, promoted a pro-inflammatory macrophage profile, and increased lymphocyte proliferation. Additionally, we demonstrated that 4Mu enhanced in vivo T cell infiltration when combined with anti-PD-1 therapy, achieving tumor growth inhibition and supporting its therapeutic potential to improve immunotherapy efficacy in lung cancer.

## 2. Results

### 2.1. Hypoxia-Related Metabolic Genes Are Upregulated in Lung Cancer Patients and Correlate with Immune Suppression

Analysis of The Cancer Genome Atlas (TCGA) datasets first revealed that NSCLC patients with stage I–IV adenocarcinoma (LUAD; n = 507) and squamous cell carcinoma (LUSC; n = 502) had significantly higher expression of *HIF1α*, *CA9* (CAIX), *SLC16A3* (MCT4), *SLC2A1* (glucose receptor-1; GLUT1), and pyruvate dehydrogenase kinase 1 (PDK1) in tumor tissues compared with adjacent normal tissues (*p* < 0.001; Figure 1A,B). In both histological subtypes, we observed enrichment of glycolytic gene expression that strongly correlated with hypoxia signatures. LUAD tumors showed a more heterogeneous distribution of glycolysis scores, whereas LUSC tumors exhibited overall higher baseline glycolysis and hypoxia levels (Figure 1A,B). In addition, in LUAD patients, high *CA9* expression was associated with reduced OS (*p* < 0.001; Figure 1C). We observed that *CA9* expression strongly correlated with metabolic reprogramming factors such as *HIF-1α*, *SLC2A1*, *SLC16A3*, *LDHA*, and *PDK1*, a key anaerobic glycolysis enzyme, which inactivates the pyruvate dehydrogenase complex enzyme (PDH), blocking the entry of pyruvate into the mitochondria toward lactate production [14] (Figure 1D). Significantly, *CA9* expression is also strongly correlated with the *CD274* gene (PD-L1; *p* < 0.001; Figure 1D). Furthermore, the LUAD heatmap showed a correlation between *CA9* expression and *FOXP3*, the distinctive Treg transcription factor (*p* < 0.05), the immunosuppressive cytokine Transforming Growth Factor beta (*TGF-β*; *p* < 0.01), the exhaustion marker Lymphocyte Activation gen 3 (*LAG-3*; *p* < 0.001), and *HAVCR2*, also known as T cell immunoglobulin and mucin domain 3 molecule (*TIM-3*; *p* < 0.01; Figure 1D). These findings support a link between hypoxia-driven metabolic reprogramming and immune suppression in the TME. We also analyzed patients with LUSC and observed that they generally exhibited higher glycolysis and hypoxia levels than LUAD, but correlations with immune checkpoints were more modest. Glycolysis-high LUSC tumors were enriched in M2-like macrophage signatures, but *CD274* induction was less pronounced than in LUAD (Figure 1E).

Currently, the expression of PD-L1 in tumor cells is the biomarker used clinically to select the treatment for NSCLC patients without driver mutations, and it correlates with better response rates, particularly when it is higher than 50% [15]. However, even in this group of patients, primary resistance is observed, highlighting the need to identify new predictors and strategies that enhance the efficacy of immunotherapy. We analyzed the expression levels of *HIF1α*, *SLC16A3*, *SLC2A1*, *LDHA*, *CA9*, and *PDK1* in early (stages I–II) and advanced (stages IIIb–IV) NSCLC patients with low or high *CD274* expression (Figure 2). In early LUAD and LUSC, expression levels of hypoxia/glycolysis-related genes were moderate and showed modest positive correlations with *CD274*. Notably, in early LUAD, *CA9* expression already trended with higher *CD274*, suggesting the emergence of a microenvironment inclined to immune evasion (Figure 2A).

In advanced disease, the association between glycolytic/hypoxic gene expression and CD274 was considerably stronger. Advanced LUAD tumors with high *CA9*, *LDHA*, and *PDK1* expression consistently displayed elevated *CD274* expression, with higher correlation coefficients than in early stages. In LUSC, although baseline glycolysis and hypoxia signatures were higher, the relationship with PD-L1 was less consistent (Figure 2B).

### 2.2. Exploratory Analysis of Prognostic Impact of PD-L1 and Hypoxia-Related Total Lesion Glycolysis

We conducted a retrospective single-center cohort study including 32 patients with advanced NSCLC (stage III n = 6; IV n = 26), treated with ICIs between 2018 and 2023. Histology comprised adenocarcinoma (n = 23), squamous cell carcinoma (n = 7), and undifferentiated tumors (n = 2). This study has limitations inherent to its retrospective design and small sample size. Median age at treatment initiation was 67 years, 57% were male, and 78% had a smoking history. Most patients (91%) received ICIs in the first-line setting, and 67% received concurrent chemotherapy.

Patients were categorized according to PD-L1 expression (positive/negative) and a primary tumor and hypoxia-associated parameter calculated as Total Lesion Glycolysis [16,17] (TLG; high/low, dichotomized at the median given the limited sample size and solely for exploratory purposes), generating four prognostic groups: PD-L1 negative/TLG low; PD-L1 negative/TLG high; PD-L1 positive/TLG low; PD-L1 positive/TLG high. There was no significant association between PD-L1 expression and TLG (*p* = 0.91), nor in objective response rates across TLG groups (*p* = 0.89). However, patients with low TLG had markedly longer PFS compared with those with high TLG: median PFS was 17.7 months (95% CI 1.7–32.2) vs. 5.5 months (95% CI 0.5–11.7), HR 0.42 (*p* = 0.05; univariate Cox model; Figure 2C). Similarly, OS was significantly improved in the low-TLG group: median OS was 40.4 months (95% CI 13–67) vs. 10.9 months (95% CI 0.9–20.9) in the TLG-high group, HR 0.31 (*p* = 0.01; univariate Cox model; Figure 2C). While PD-L1 expression alone was not associated with outcomes, the combined analysis of PD-L1 and TLG identified distinct prognostic subgroups. Patients with low TLG and positive PD-L1 expression experienced the most favorable outcomes, supporting the potential role of TLG as a complementary biomarker in advanced NSCLC (Figure 2D).

### 2.3. Exposure of Lung Cancer Cells to Hypoxia Induces a Hostile Microenvironment for Immune Cells

We next used experimental models to validate the observed data from NSCLC patients, which allow the evaluation of therapeutic strategies targeting metabolic variations and the impact on immune evasion. Then, we tested the effect of hypoxia-induced metabolic reprogramming in human lung adenocarcinoma (A549) and murine (Lewis lung carcinoma, LLC) cell lines. HeLa cells transfected with the 5HRE/green fluorescent protein (GFP) reporter plasmid were used as a reference system to evaluate hypoxia-dependent responses (Appendix A). Hypoxic exposure did not compromise tumor cell viability, as both LLC and A549 cells remained viable after 24 and 48 h in low-oxygen conditions (Figure 3A). Instead, hypoxia induced a reduced mitochondrial activity (Figure 3B, left) and marked metabolic adaptations. We focused on analyzing CAIX expression, observing an upregulation of gene expression after 48 h of cell culture in LLC cells (*p* < 0.001; Figure 3B, middle). Notably, these changes were accompanied by decreased extracellular pH, and increased lactate release (*p* < 0.05; Figure 3B, right). Similar results regarding CAIX expression, extracellular pH, and lactate were obtained in A549 cells (Figure 3C). Additionally, we observed changes in Vascular Endothelial Growth Factor (VEGF), MCT-4, and in PD-L1 expression in LLC (Appendix A).

Coumarins have been identified as a novel class of carbonic anhydrase (CA, EC 4.2.1.1) inhibitors. Their inhibitory mechanism involves hydrolysis of the lactone ring catalyzed by the intrinsic esterase activity of CA, generating derivatives that occupy a region of the active site not targeted by other inhibitor classes. This unique binding mode accounts for their remarkable selectivity as CA inhibitors [18,19]. We have previously reported the TME modulation by coumarin 4Mu. Here, we aimed to analyze whether 4Mu could reverse the metabolic alterations induced by the increase in CAIX activity related to hypoxia. In both LLC and A549 cells, 4Mu treatment normalized extracellular pH, reduced lactate accumulation, and suppressed CAIX upregulation at both 24 h and 48 h (Figure 3B). These effects suggest that 4Mu interferes with the establishment of an acidic, immunosuppressive microenvironment by attenuating hypoxia-driven glycolysis and CAIX expression. Furthermore, we verified that the addition of the coumarin 4Mu to the culture medium did not intrinsically modify extracellular pH (Appendix A).

Classical carbonic anhydrase inhibitors have been demonstrated to exert potent activity against CAIX, resulting in a suppression of cancer cell proliferation in both in vitro and in vivo models [20,21,22]. Representative compounds of this class include acetazolamide (Az), dichlorphenamide, and dorzolamide [23]. LLC cells were incubated with increasing concentrations of Az under normoxia and hypoxia, and CAIX gene expression was assessed at 48 h (Appendix A). We observed that hypoxic culture conditions induced a marked upregulation of CAIX expression (*p* < 0.001). However, Az reduced CAIX mRNA levels, confirming its modulatory effect on gene expression even in normoxia (*p* < 0.01 and *p* < 0.001). Nevertheless, this reduction did not result in measurable changes in CAIX functionality, as assessed by extracellular pH at 48 h, in contrast with the effects observed with 4Mu (Appendix A). Possibly longer exposure times or alternative dosing strategies may be required to achieve a measurable impact on CAIX functionality using Az. On the other hand, it has been reported that Az activates acid-sensing ion channels inducing acidification of extracellular environment [24,25].

Extracellular acidification due to lactate accumulation has been proposed as a hallmark of cancer, but it has also been assumed that lactate is only a waste product of this metabolic process. However, lactate may play an indispensable role in shaping immune cell function, regulating their metabolism and inhibiting their activation [26]. In this sense, murine J774 macrophages exposure to conditioned media from hypoxic tumor cells showed a protumor phenotype, promoted by the altered gene expression of Arginase-1 (Arg-1), inducible Nitric Oxide synthase (iNOS), and the M2 marker CD206, as well as anti-inflammatory interleukin 10 (IL-10) production (Figure 4A; conditioned media from hypoxic tumor cells (CH) vs. conditioned media from normoxic tumor cells (CN)).

Additionally, proliferation assays using CFSE labeling demonstrated that conditioned media from hypoxic tumor cells impaired antigen-specific splenocyte proliferation after 5 days of stimulation (*p* < 0.01), highlighting the immunosuppressive capacity of the hypoxic TME (Figure 4B). In contrast, conditioned media from 4Mu-treated tumor cells reversed macrophage polarization toward an inflammatory profile, and rescued splenocyte proliferation (*p* < 0.01; conditioned media from hypoxic 4Mu-treated tumor cells (HMu) vs. conditioned media from normoxic 4Mu-treated tumor cells (NMu)), demonstrating that 4Mu alleviates metabolic and immune dysfunction induced by hypoxia. These findings reveal that tumor-derived soluble factors under hypoxia exert direct inhibitory effects on immune cell activity, whereas 4Mu treatment mitigates this immunosuppressive communication.

### 2.4. 4Mu Modulates the TME and Enhances Antitumor Immunity In Vivo

Next, we aimed to test the effects observed in vitro using murine lung cancer-bearing mice. To this aim, LLC cells were subcutaneously injected into syngeneic C57 mice (day 0; n = 10). On day 8, animals with tumors reaching 80 mm^3^ in size, were orally treated with 200 mg/kg/day of 4Mu (n = 5) or left untreated (control, n = 5). We recollected tumor samples at time = 0 and after 24 and 48 h post-4Mu, for CAIX immunohistochemistry. We observed that untreated tumors (control group) expressed significantly higher amounts of CAIX after 48 h of 4Mu treatment (7.5 ± 2.2% vs. 3.4 ± 1.1 of positive area control vs. 4Mu; *p* < 0.001) in comparison with tumors from 4Mu-treated mice (Figure 5A).

These results were accompanied by a significant reduction in CAIX gene expression (Figure 5B) and underscore the capability of 4Mu to decrease CAIX both in vitro and in vivo. Moreover, mice that received 4Mu showed decreased expression of transcripts associated with M2-like macrophages (Arg1, CD163, IL-10; *p* < 0.001, *p* < 0.05 and *p* < 0.01, respectively, Figure 5B), while pro-inflammatory markers IL-1β, TNFα and IFNA1 tended to be increased. Moreover, 4Mu treatment upregulated CD8 gene expression (*p* < 0.05; Figure 5C).

We examined the antitumor effects of the combined treatment in LLC tumor-bearing mice that were orally treated, on day 8 after tumor inoculation, with 200 mg/kg/day of 4Mu (Figure 6). Then, they were injected intraperitoneally 48 h later (day 10) every 3 days for a total of 3 doses, 100 µg/kg, of anti-PD1 (n = 8). Animals treated with each single agent (n = 6–8/group) or saline (n = 5–8) were also included as controls. The effects observed in mice treated with 4Mu or anti-PD-1 alone were similar to those obtained in the control group (Figure 6A). In contrast, the combination of 4Mu + anti-PD-1 consistently resulted in a marked tumor volume reduction with respect to saline and 4Mu (*p* < 0.01). In addition, the combined treatment produced a significant reduction in tumor volume when compared with animals treated with anti-PD-1 as a single agent (*p* < 0.01). The monotherapies and the combined strategy were well tolerated, with no signs of toxicity. Survival of mice receiving combined therapy was significantly increased compared to mice receiving each single treatment or saline (*p* < 0.001) (Figure 6B). The in vivo interaction between both treatments was analyzed by the fractional product method [27]. Table 1 summarizes the relative tumor volume of different groups on different time points. On day 6 after treatment (day 15 post-LLC inoculation), there was a 2.1-fold improvement in antitumor efficacy in the combination group when compared with the expected additive effect. Moreover, on days 10 and 12 after treatment, the combination showed a 1.7- and 1.3-fold increase, respectively, in the inhibition of tumor growth over an additive effect (expected fractional tumor volume). These results allow us to conclude that the 4Mu + anti-PD-1 combined therapy has a synergistic effect on tumor growth inhibition.

Finally, we assessed tumor immune infiltrates (Figure 6C,D). We observed by flow cytometry that 4Mu + anti-PD-1-treated tumors exhibited a tendency to show increased infiltration of both CD4+ and CD8+ T cells (Figure 6C). Furthermore, the analysis of lymphocytes T CD4 and T CD8 by qPCR showed a higher expression of helper and cytotoxic effector markers in mice with combined therapy vs. monotherapies (*p* < 0.001 mRNA levels of CD4+ cells in combined therapy vs. 4Mu or anti PD-1; *p* < 0.001 mRNA levels of CD8+ cells in combined therapy vs. 4Mu or *p* < 0.01 vs. anti-PD-1), supporting enhanced antitumor immunity mediated by 4Mu modulation, confirming its capacity to reprogram the TME to achieve an antitumoral effect that can improve anti-PD-1 treatment.

## 3. Discussion

The present study demonstrates that hypoxia-driven metabolic alterations are, at least in part, mediated by CAIX and play a pivotal role in shaping the immunosuppressive TME in NSCLC. This work establishes a proof-of-concept for pharmacologic modulation with 4Mu, since it can mitigate these effects while synergizing with PD-1 blockade and does not show adverse effects in a preclinical model. Our findings provide a mechanistic insight into how metabolic reprogramming is linked to immune evasion in lung tumors, suggesting that 4Mu could be exploited as an adjuvant to ICIs. Importantly, 4Mu has a known safety profile in mice [28,29,30], and its use is approved in Europe and Asia for hepatobiliary disorders [31] and is being tested for other pathologies [32,33].

Consistent with previous studies, we observed in TCGA datasets that NSCLC tumors exhibit upregulation of hypoxia-associated genes including HIF1α, GLUT1, LDHA, PDK1, MCT4, and CAIX, which collectively promote glycolytic metabolism, lactate accumulation, and extracellular acidification [34]. Elevated CAIX expression correlated with reduced OS in LUAD, reinforcing its role as a prognostic biomarker of poor clinical outcome. Importantly, CAIX expression was also associated with PD-L1 expression and other immune-inhibitory signatures (FOXP3, TGF-β, LAG-3, TIM-3), supporting the notion that metabolic stress directly contributes to immune dysfunction in the TME [35], allowing for cancer progression. These results extend prior observations in breast, renal, and colorectal cancer, where CAIX expression has been linked to immunosuppressive signaling and resistance to immunotherapy [36,37,38].

In vitro experiments confirmed that hypoxia induced CAIX upregulation, lactate release, and extracellular acidification in both human and murine lung cancer cells. The resulting tumor-conditioned media skewed macrophages toward an M2-like phenotype and impaired antigen-specific T cell proliferation, underscoring how metabolic products orchestrate immune escape. These findings align with accumulating evidence that lactate is not purely a metabolic waste product but also a signaling molecule that shapes macrophage polarization and T cell function to modify the tumor immune milieu into an immunosuppressive one [11,39,40].

Here, we demonstrated that 4Mu effectively counteracted hypoxia-induced CAIX expression, restored extracellular pH, and reduced lactate secretion. Regarding 4Mu’s effects on immune cells, it was able to reverse macrophage polarization and rescue splenocyte proliferation, thereby alleviating hypoxia-driven immune suppression. Coumarins, including 4Mu, have been reported as selective carbonic anhydrase inhibitors with unique binding properties distinct from sulfonamide-based inhibitors [41,42]. Moreover, 4Mu has the advantage, compared to other CAIX inhibitors, of already being approved for human use [31]. Our data expands these observations, demonstrating that 4Mu not only modulates CAIX activity but also remodels the TME toward an immunostimulatory state.

In vivo, oral administration of 4Mu decreased CAIX expression, reduced M2-like macrophage markers, and increased CD8+ T cell infiltration, corroborating its immunomodulatory properties. Notably, 4Mu alone does not exert antitumor effects, but when combined with anti-PD-1 therapy, it produces a synergistic inhibition of tumor growth and significantly improves survival in a preclinical model that resembles LUAD, the most frequent NSCLC histological subtype. The enhanced infiltration of CD4+ and CD8+ T cells in the combination group highlights that 4Mu potentiates checkpoint blockade efficacy by reprogramming the metabolic landscape of tumors. This is highly relevant because the LLC model is reported to be a poor responder to immunotherapy [43]. On the other hand, despite the success of ICIs in NSCLC, objective response rates remain around 20–30%, and primary or acquired resistance remains a major challenge [44,45].

Our results parallel findings from preclinical studies targeting lactate metabolism, acidosis, or hypoxia-inducible pathways to enhance immunotherapy outcomes. For example, inhibition of LDHA or blockade of lactate transporters has been shown to restore T cell function and augment ICI efficacy in murine models [46,47]. However, these strategies often have limitations related to toxicity or metabolic compensation. The advantage of 4Mu is related to its oral bioavailability, tolerability, and dual capacity to modulate both tumor metabolism and immune responses, making it an attractive candidate for further clinical studies. Additionally, while we focused on CAIX as a primary target of 4Mu, other hypoxia-related pathways such as CAXII, LDHA, and VEGF may also be modulated and should be explored in future studies.

On the other hand, our clinical cohort analysis showed that patients with low tumor glycolytic activity, measured as TLG, experienced the most favorable outcomes. The use of the median as a cut-off for TLG was exploratory, underscoring the need for larger cohorts to define specific thresholds and validate TLG as a biomarker before its prognostic or predictive role can be more accurately established.

In conclusion, our findings identify CAIX-driven metabolic reprogramming as a central mechanism of immune evasion in NSCLC and demonstrate that pharmacologic modulation with 4Mu mitigates hypoxia-induced acidosis, remodels the TME, and synergizes with PD-1 blockade to obtain an antitumor effect. These results support further investigation of 4Mu as a metabolic adjuvant to immunotherapy and highlight the value of integrating metabolic biomarkers such as TLG with PD-L1 for improved prognostic and predictive assessment in lung cancer.

## 4. Materials and Methods

### 4.1. TCGA Database Analysis

Gene expression (RNA sequencing [RNA-Seq]) data for the indicated genes were obtained from The Cancer Genome Atlas (TCGA) for patients with non-small cell lung cancer (NSCLC). Datasets included lung adenocarcinoma (LUAD; n = 507) and lung squamous cell carcinoma (LUSC; n = 502), which were analyzed separately. For tumor (T) versus non-tumoral adjacent tissue (NTAT) comparisons, paired samples were used to calculate individual fold changes. Correlation analyses between gene expression levels were performed by linear regression using RStudio 2025.05.1+513 software, and correlation coefficients (R) > 0.2 were considered positive. Kaplan–Meier survival analyses for specific genes were conducted using TCGA clinical data. Data supporting the reported results can be accessed at https://xena.ucsc.edu/.

### 4.2. Cells and Cell Culture

Murine Lewis lung carcinoma (LLC, ATCC CRL-1642 kindly transferred by Dra. Ada Blinder and Dr. Gabriel Rabinobich, IBYME-CONICET, Buenos Aires, Argentina), murine macrophage J774, and human A549 lung carcinoma (ATCC CCL-185) cells were cultured in Dulbecco’s Modified Eagle Medium (DMEM; Serendipia Lab., Vedia, Buenos Aires, Argentina) or RPMI-1640 (Gibco, Invitrogen Argentina, Buenos Aires, Argentina) according to cell line requirements, supplemented with 10% fetal bovine serum (Natocor, Córdoba, Argentina) and 1% penicillin–streptomycin (Gibco). Cells were maintained at 37 °C in a humidified incubator with 5% CO_2_. For hypoxia experiments, cells were incubated at 1% O_2_, 5% CO_2_, and 94% N_2_ in a Modular Incubator Chamber (MIC-101; Embrient Inc., San Diego, CA, USA). Cells were treated with 4-methylumbelliferone (4Mu; Sigma-Aldrich, San Louis, MO, USA) at 125, 250, or 500 µM, or with vehicle (Hank’s balanced salt solution, HBSS, Sigma-Aldrich, San Louis, MO, USA), for 24, 48 and 72 h. In parallel, acetazolamide (Az; ACETA^®^, Laboratorios Pablo Cassara, Buenos Aires, Argentina) was used at concentrations of 1, 2.5, and 5 µM [48] for 48 h under normoxic or hypoxic conditions.

#### 4.2.1. Cell Viability Assessment and Metabolic Activity Assay In Vitro

Cell viability and morphological changes were assessed by acridine orange–ethidium bromide mixture staining (Sigma). Single-cell suspensions were stained with 10 μg/mL of the mixture, and cells were visualized under a fluorescence microscope. At least 100 cells were counted from 4 independent experiments, and the percentage of viable cells was determined and visualized by fluorescence microscopy (Nikon Eclipse E800, Nikon, Buenos Aires, Argentina). Metabolic activity was measured using the colorimetric MTT assay (Invitrogen). Briefly, 5 × 10^3^ cells (A549 and LLC) seeded onto 96-well plates in a final volume of 100 μL per well and incubated for 24 and 48 h in normoxia or hypoxia. Then, the culture medium was replaced with 100 μL of 5 mg/mL 3-(4,5-dime-thylthiazol-3-yl)-2,5-diphenyl tetrazolium bromide. Four hours later, the formazan dye was solubilized and read at 490 nm optical density using a spectrophotometer. Each assay was performed 4 times in quadruplicate.

#### 4.2.2. Conditioned Medium

pH and lactate concentration: conditioned media from LLC cells were cultured under different conditions—normoxia, normoxia + 4Mu (250 µM), hypoxia, and hypoxia + 4Mu—were collected. Lactate concentration was measured using the Lactate Assay Kit (Wiener, 1999795, Rosario, Argentina), following the manufacturer’s instructions. The pH of conditioned media was determined using nanoelectrodes [49].

### 4.3. Macrophage Polarization Assay

Murine J774 macrophages were incubated for 24 h with the above LLC-conditioned media (normoxia, normoxia + 4Mu, hypoxia, hypoxia + 4Mu). After treatment, cells were harvested for downstream analyses to evaluate their pro- or anti-inflammatory profile.

### 4.4. Splenocyte Proliferation Assay

C57BL/6 mice were subcutaneously inoculated with 2 × 10^6^ LLC cells. Fourteen days later, spleens were collected under aseptic conditions, and single-cell suspensions of splenocytes were prepared. Cells were labeled with CFSE (Invitrogen, C34554, Waltham, MA, USA)), a fluorescent dye that covalently binds to intracellular proteins. Upon each round of cell division, CFSE fluorescence intensity is reduced by half, allowing the quantification of lymphocyte proliferation by flow cytometry. Splenocytes were cultured for 5 days in the presence of LLC-conditioned media from the four experimental conditions (normoxia, normoxia + 4Mu, hypoxia, hypoxia + 4Mu), and proliferation was subsequently evaluated by CFSE dilution analysis.

### 4.5. RNA Isolation and Quantitative PCR Analysis

Total RNA was extracted using TRI Reagent (Sigma-Aldrich, St. Louis, MO, USA) following the manufacturer’s instructions. For cDNA synthesis, 1 µg of RNA was reverse-transcribed with 200 U of SuperScript II Reverse Transcriptase (Invitrogen, Carlsbad, CA, USA) using 500 ng oligo (dT) primers. Quantitative real-time PCR (qRT-PCR) was performed using a Stratagene Mx3005p thermocycler (Stratagene, La Jolla, CA, USA) and SYBR Green chemistry (Invitrogen, Carlsbad, CA, USA). Gene expression was analyzed for ARG1, CAIX, CD163, CD206, CD4, CD8, GLUT1, IL1b, IL10, iNOS, MCT4, PDL1, TGFb, TNFa, using specific primers listed in Table 2 (m = mouse, h = human). Amplification conditions were as follows: initial denaturation at 95 °C for 10 min, followed by 45 cycles of 95 °C for 30 s, 60 °C for 30 s, and 72 °C for 1 min. β-actin (ACTb) was used as the internal housekeeping control.

Relative expression levels were calculated using the 2^−ΔΔCt^ method, setting untreated cells (control) as the reference condition. A non-template control (NTC) was included in each run, and all measurements were performed in triplicate in three independent experiments.

### 4.6. In Vivo and Ex Vivo Experiments

#### 4.6.1. Animals

Female C57BL/6 mice (6–8 weeks old) were housed in the Animal Resources Facility under standard conditions, in accordance with the NIH guidelines for the ethical use of animals in research.

#### 4.6.2. Evaluation of Tumor CAIX Expression

Syngeneic C57BL/6 mice were subcutaneously injected into the flank with 2 × 10^6^ LLC cells (day 0). On day 8, when tumors reached ~80 mm^3^ as measured with a caliper, animals were assigned to two groups, (1) control (water) and (2) 4Mu (200 mg/kg/day, administered orally during 48 h), n = 5 animals/group. Subsets of mice were euthanized at time= 0 or 24 and 48 h after treatment initiation for tissue collection. Tumors were excised, fixed in neutral-buffered formalin, and paraffin-embedded. Sections (5 µm) were prepared for histology and immunohistochemistry (IHC). The adjacent pieces from samples collected at 48 h were snap-frozen for RNA extraction and quantitative PCR Analysis. The experiment was carried out 2 times.

#### 4.6.3. Immunohistochemical Analysis for CAIX

For immunohistochemistry, sections were deparaffinized, rehydrated, and subjected to antigen retrieval. Endogenous peroxidase activity was blocked, and slides were incubated overnight at 4 °C with a primary antibody against CAIX (Anti-Carbonic Anhydrase IX Antibody, Rabbit Polyclonal Cat: 50660-T24, Sino Biological, Houston, TX,, USA). After washing, sections were incubated with a biotinylated secondary antibody (Anti-Rabbit IgG (whole molecule)–Biotin antibody B8895, Sigma-Aldrich) for 1 h at room temperature, followed by streptavidin–HRP incubation. Immunoreactivity was visualized using DAB as chromogen. Stained slides were scanned (Leica™, Wetzlar, Germany ) and analyzed to determine the percentage of CAIX-positive areas using ImageJ software version 1.54g, National Institutes of Health, Bethesda, MD, USA, NIH.

#### 4.6.4. Combination Therapy Studies In Vivo

C57BL/6 mice were subcutaneously inoculated with 2 × 10^6^ LLC cells into the left flank (day 0). When tumors became palpable and reached a volume of approximately 65–70 mm^3^, as measured with a caliper, animals were randomly assigned into four experimental groups (n = 5–8 per group): (1) control; (2) 4-methylumbelliferone (4Mu, 200 mg/kg/mouse, administered orally); (3) anti-PD-1 antibody (administered every 3 days for a total of three doses); and (4) combined treatment with 4Mu and anti-PD-1 following a reported schedule [50]. Tumor volume was measured with a caliper 3 times per week and tumor volume was calculated as follows: π/6 × larger diameter × (smaller diameter)^2^. Weight was registered once a week, and the maximal tumor size permitted by our Animal Care Committee was not exceeded. All procedures were conducted in accordance with our Animal Care Committee guidelines, based on Guide for the Care and Use of Laboratory Animals (NIH, Bethesda, MD, USA) and the AVMA Guidelines for the Euthanasia of Animals. This study complies with the ARRIVE guidelines for reporting in vivo experiments. The experiment was carried out 2 times.

#### 4.6.5. Flow Cytometry

On day 21 after LLC inoculation, tumor tissues from the control (n = 3), 4Mu (n = 3), an-ti-PD-1 (n = 3), and combined treatment with 4Mu and anti-PD-1 (n = 3) groups were excised, mechanically dissociated, and enzymatically digested to obtain single-cell suspensions. After washing and filtering, cells were incubated with fluorochrome-conjugated mono-clonal antibodies against CD4 (rat anti-mouse CD4-PE, BD Biosciences, Franklin Lakes, NJ, USA) and CD8 (rat anti-mouse CD8-Alexa Fluor 488, BD Biosciences, Franklin Lakes, NJ, USA) for 40 min at 4 °C in the dark. Stained cells were washed, resuspended in PBS, and subjected to flow cytometry analysis using a BD Accuri™ C6 Plus Personal Flow Cytometer (BD Biosciences, Franklin Lakes, NJ, USA). Data acquisition and analysis were performed with BD Accuri C6 software version 1.0.23.1.

#### 4.6.6. Quantitative PCR Analysis for CD4 and CD8 Markers in Tumor Samples After Combined Treatment

On day 21 after LLC inoculation, tumor tissues from control (n = 2), 4Mu (n = 3), anti-PD-1 (n = 3), and combined treatment with 4Mu and anti-PD-1 (n = 4) were excised; RNA isolation was assessed and gene expression for CD4 and CD8 was analyzed as we described above.

#### 4.6.7. Retrospective Patient Study Design

A retrospective cohort study was conducted including patients with advanced NSCLC without driver mutations who received ICI therapy. All patients underwent baseline staging with 18FluorDeoxiGlucose (FDG) Positron Emission Tomography (PET)/Computed Tomography (CT) prior to treatment initiation between 2018 and December 2023. This study was carried out exclusively at Hospital Universitario Austral and was an observational retrospective study. Eligible patients were required to fulfil the following criteria: histologically confirmed advanced NSCLC (stage IIIb/c or IV); treatment-naïve patients who received systemic therapy with ICIs in the first- or second-line setting; candidates for immunotherapy directed against the immune checkpoints PD-1, PD-L1, or CTLA-4; negative molecular testing for driver mutations such as Epidermal Growth Factor Receptor (EGFR), Anaplastic Lymphoma Kinase (ALK), and Oncogen ros1 (ROS1); FDG PET/CT performed within 90 days prior to initiation of ICI therapy; and primary tumor size ≥ 1 cm. Exclusion criteria were patients with missing data or lost to follow-up and with non-evaluable disease according to the Response Evaluation Criteria in Solid Tumors (RECIST) criteria. We measured demographic, clinical, and survival variables. Data was retrieved from electronic medical records and entered into a study-specific database. Collected variables included sex, age at treatment initiation, Eastern Cooperative Oncology Group (ECOG) performance status, smoking history, histology, PD-L1 expression by immunohistochemistry (all clones accepted), and clinical Tumor Staging System (TNM) staging [51]. PFS was defined as the time from treatment initiation to disease progression according to RECIST 1.1 or death, and OS as the time from treatment initiation to death. Patients lost to follow-up were censored at the last recorded visit. Treatment response was assessed according to RECIST 1.1 and clinical evaluation by the treating physician (C.R.P.; J.N.M.; R.A.). Response categories included complete response (CR), partial response (PR), stable disease (SD), and progressive disease (PD). Objective response rate (ORR) was defined as CR+PR, and disease control rate (DCR) as CR+PR+SD. When CT evaluation was not feasible due to rapid progression or clinical deterioration, response was assessed clinically and with laboratory data only. FDG PET/CT images were retrieved from the institution, anonymized, and reviewed by two board-certified nuclear medicine physicians blinded to patient outcomes. Radiomic features were extracted from PET images using LIFEx software version 7.8.4. Measurements included primary tumor characteristics (tumor size, standardized uptake value (SUV) SUV max, SUV mean, Metabolic Tumor Volume (MTV), Total Lesion Glycolysis (TLG)), and heterogeneity histogram (AUC-CSH) and whole-body tumor burden (SUV max, SUV meant, MTV, and TLG). Volumes of interest (VOIs) were delineated semi-automatically using a threshold of 40% of SUV max (except for heterogeneity, where SUV max ≥ 3 was applied). Radiomic variables such as TLG were dichotomized as “high” or “low” based on the median value, as distributions were non-normal and no reference thresholds are available in the literature.

### 4.7. Statistical Analysis

One-way and two-way ANOVAs with the corresponding post hoc tests such as Kruskal–Wallis or Tukey and Mann–Whitney or Student’s *t*-test (InStat, GraphPad 8 Software) were used to examine the statistical differences between the groups. Mouse survival was analyzed by a Kaplan–Meier curve and Mantel–Cox log rank test. *p* < 0.05 was considered statistically significant.

## Figures and Tables

**Figure 1 ijms-26-10427-f001:**
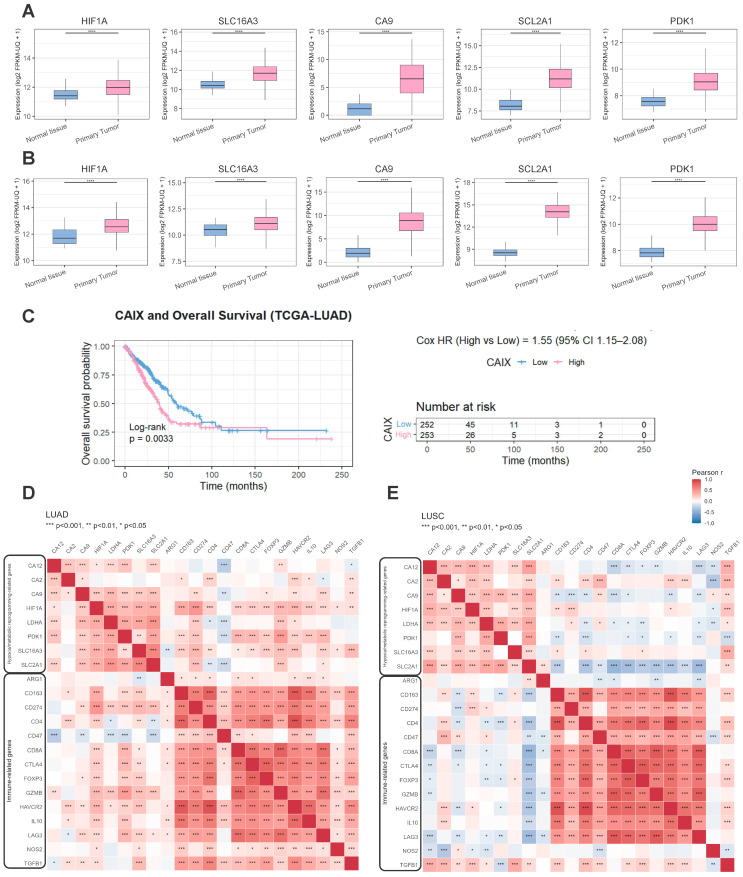
TCGA analysis of hypoxia and metabolism-related genes in NSCLC. (**A**,**B**) Gene expression analysis of *HIF1α*, *CA9*, *SLC16A3, PDK1*, and *SLC2A1* in paired samples of tumor tissue versus adjacent non-tumoral tissue from LUAD (n = 507) and LUSC (n = 502) patients in TCGA. **** *p* < 0.0001 versus normal tissue (paired *t*-test). (**C**) Kaplan–Meier overall survival curves for LUAD patients stratified by *CA9* expression, showing reduced survival in patients with high *CA9* expression (*p* < 0.001, log-rank test). (**D**,**E**) Heatmaps of Pearson correlations among hypoxia/metabolic reprogramming-related and immune-related genes in LUAD (n = 511) and LUSC (n = 506) cohorts. The color scale indicates the strength and direction of the correlation (red: positive; blue: negative). Asterisks denote statistical significance (* *p* < 0.05, ** *p* < 0.01, *** *p* < 0.001); two-tailed significance test for Pearson correlation.

**Figure 2 ijms-26-10427-f002:**
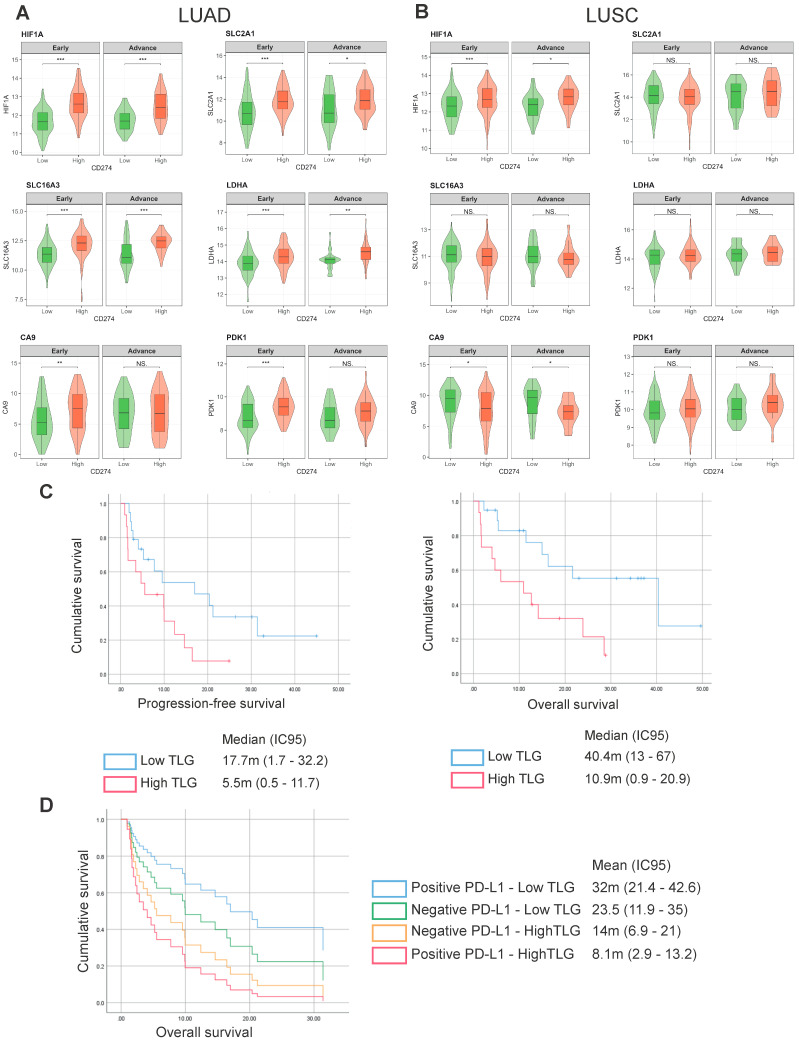
Association of hypoxia/glycolysis-related genes with CD274 expression in LUAD and LUSC patients. Violin plots show the expression of *HIF1α*, *SLC2A1*, *SLC16A3*, *LDHA*, *CA9*, and *PDK1* in LUAD (**A**) and LUSC (**B**) tumors, stratified by disease stage (early vs. advanced) and *CD274* (PD-L1) expression (low vs. high). In early-stage LUAD and LUSC, moderate expression levels of hypoxia/glycolysis-related genes showed only modest associations with *CD274*. However, in early-stage LUAD, *CA9* expression already trended with higher *CD274* levels. In advanced disease, the correlation between hypoxia/glycolysis-related genes and *CD274* expression was considerably stronger, particularly in LUAD, where tumors with high *CA9*, *LDHA*, and *PDK1* expression displayed elevated *CD274*. Asterisks denote statistical significance (* *p* < 0.05, ** *p* < 0.01, *** *p* < 0.001), Mann–Whitney U test. (**C**) Progression-free survival (left) and overall survival (right) according to TLG (*p* = 0.05 and *p* = 0.01, respectively; univariate Cox model); (**D**) overall survival according to PD-L1 and TLG expression (*p* = 0.18, univariate).

**Figure 3 ijms-26-10427-f003:**
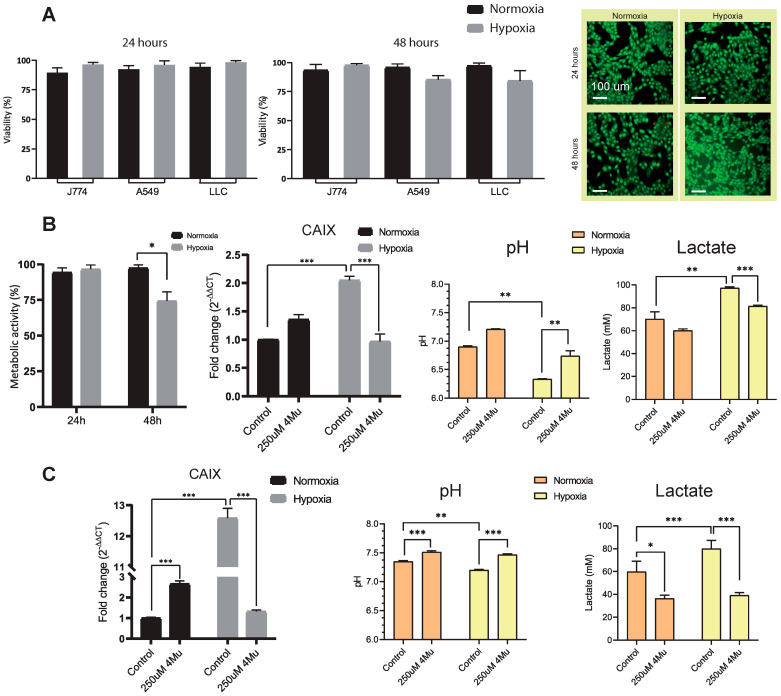
Effect of hypoxia on cell viability and metabolic reprogramming. (**A**) Cell viability of murine J774 and LLC cells and human A549 lung carcinoma cells cultured under normoxia or hypoxia for 24 and 48 h, showing no significant reduction in viability under hypoxic conditions, unpaired *t*-test. (**B**) Metabolic adaptations induced by hypoxia. Mitochondrial activity at 24 and 48h in LLC cultured under normoxia or hypoxia (**left**). Relative CAIX mRNA expression was significantly upregulated in LLC (**middle**). Hypoxic exposure also increased lactate concentration in the culture supernatant and decreased extracellular pH compared with normoxia (**right**). (**C**) Similar results were obtained in A549 cells. Data are shown as mean ± SEM of two independent experiments * *p* < 0.05, ** *p* < 0.01 and *** *p* < 0.001 versus normoxia; two-way ANOVA with Tukey’s multiple-comparison test.

**Figure 4 ijms-26-10427-f004:**
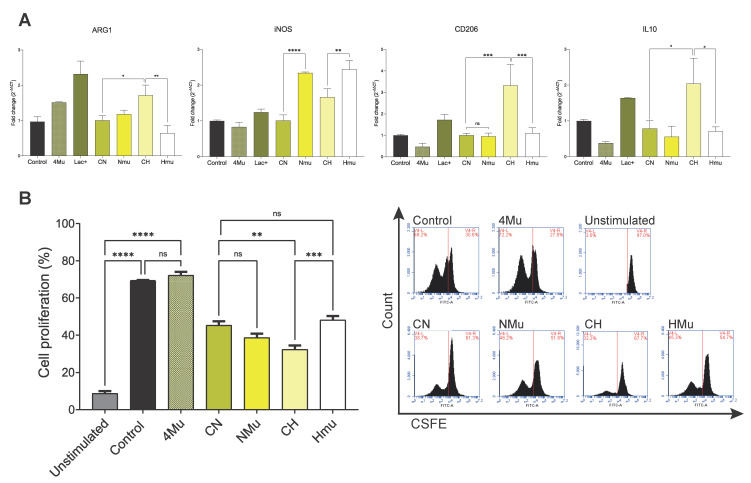
4Mu reversed hypoxia-induced lactate accumulation immunosuppressive effects on macrophages and specific lymphocytes. (**A**) Gene expression analysis of J774 macrophages exposed to conditioned media from LLC tumor cells cultured under different conditions: normoxia (CN); hypoxia (CH). Conditioned media from hypoxic tumor cells (CH) induced a protumoral profile characterized by increased expression of Arg1, iNOS, CD206, and IL-10, whereas conditioned media from 4Mu-treated cells (HMu) shifted polarization toward an inflammatory phenotype. (* *p* < 0.05, ** *p* < 0.01, *** *p* < 0.001, **** *p* < 0.0001), One-way ANOVA with Tukey’s multiple comparisons test. (**B**) CFSE-based proliferation assays of splenocytes incubated for 5 days with the indicated conditioned media. Hypoxic tumor supernatants significantly impaired splenocyte proliferation, while 4Mu treatment restored antigen-specific proliferation. Data are shown as mean ± SEM; ** *p* < 0.01, *** *p* < 0.001, **** *p* < 0.0001, One-way ANOVA with Tukey’s multiple comparisons test.

**Figure 5 ijms-26-10427-f005:**
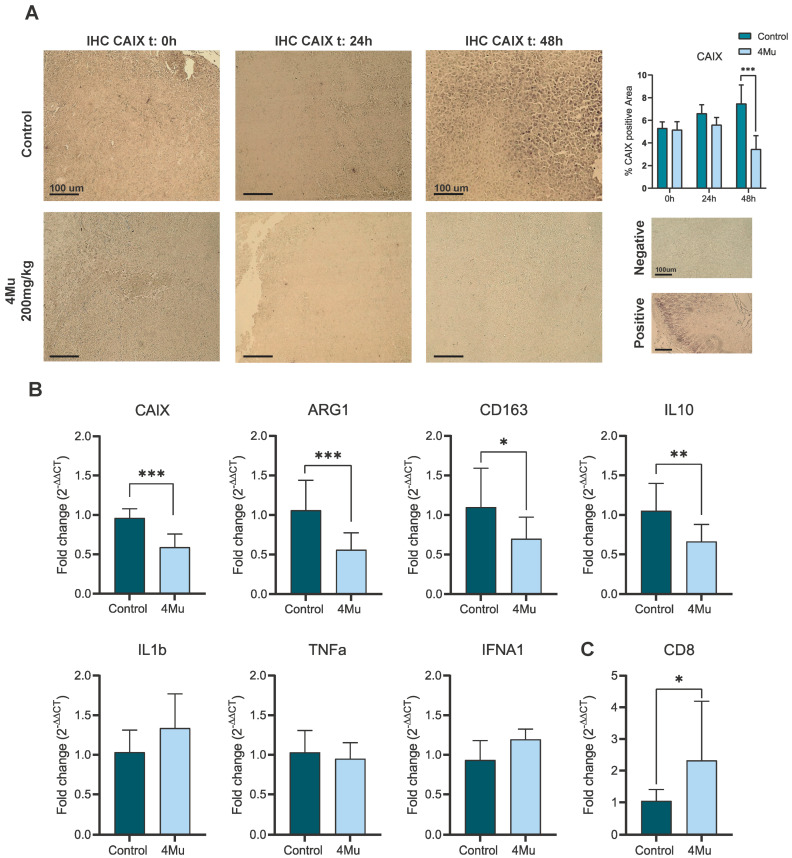
4Mu modulates the tumor microenvironment and enhances antitumor immunity in vivo. (**A**) Representative immunohistochemistry analysis of CAIX in tumor sections from LLC-bearing mice treated with vehicle (control) or 4Mu (200 mg/kg/day, oral gavage for 48 h). Quantification of CAIX-positive area by densitometry was determined at start of treatment, 24 and 48 h. A significant reduction upon 48 h 4Mu treatment was observed (*** *p* < 0.001), unpaired *t*-test. Magnification of tumor regions (10×), scale bar = 100 μm. (**B**) qRT-PCR analysis of tumor tissues revealed decreased CAIX mRNA levels in 4Mu-treated mice at 48 h, together with downregulation of M2 macrophage-associated transcripts Arg1, CD163, and IL-10 (* *p* < 0.05, ** *p* < 0.01, *** *p* < 0.001, respectively, unpaired *t*-test) and a trend toward increased pro-inflammatory markers IL-1β, TNFα, and IFNA1. (**C**) CD8 expression was significantly upregulated in tumors from 4Mu-treated mice (* *p* < 0.05), unpaired *t*-test. Data are presented as mean ± SEM.

**Figure 6 ijms-26-10427-f006:**
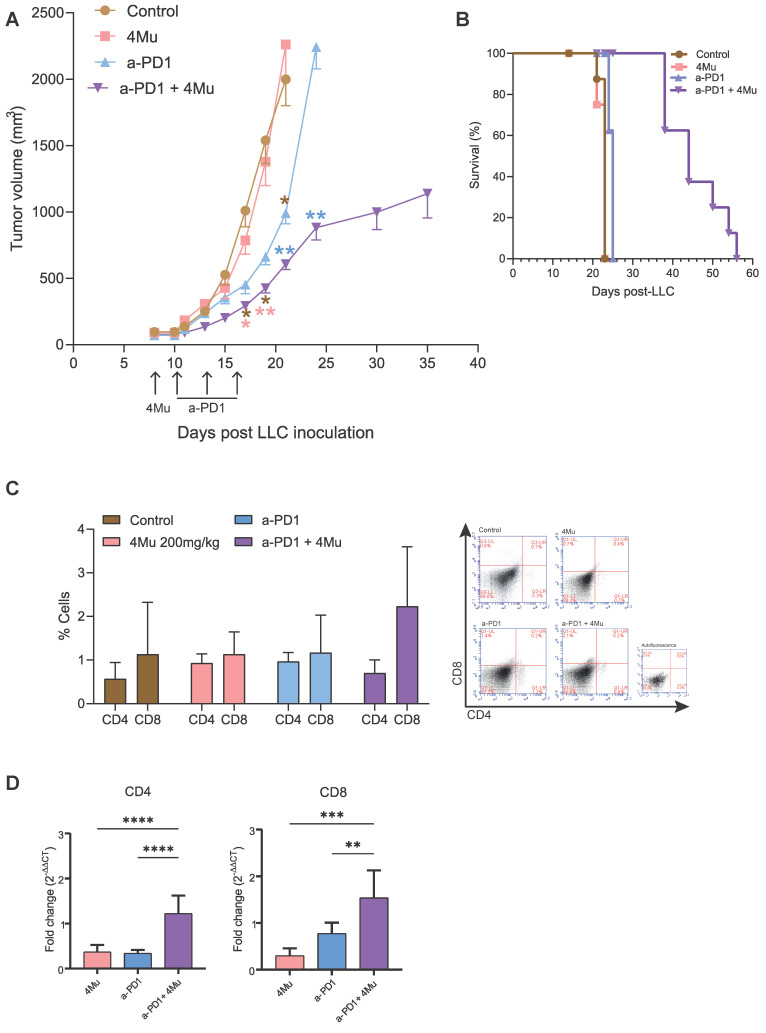
Combined treatment with 4Mu and anti-PD-1 enhances antitumor efficacy and immune infiltration in vivo. (**A**) Tumor growth curves of LLC-bearing mice treated with vehicle (control n = 8), 4Mu (200 mg/kg n = 8), anti-PD-1 (n = 8), or the combination 4Mu + anti-PD-1 (n = 8). Combination therapy resulted in a significant reduction in tumor volume compared with control, 4Mu, or anti-PD-1 alone (* *p* < 0.05, ** *p* < 0.01), ANOVA (days 17, 19 and 21) or Student’s T test (day 24) post LLC-administration. (**B**) Kaplan–Meier survival curves of treated mice. Survival was significantly prolonged in the combination group compared with single-agent treatments or control, log-rank (Mantel–Cox) test. (**C**) Flow cytometry analysis of tumor-infiltrating lymphocytes. Data show mean ± SEM from triplicates. Representative flow cytometry plots are shown. (**D**) Tumors from mice treated with the combined regimen showed higher expression of CD8 and CD4 mRNA levels measured by qPCR (**** *p* < 0.0001 mRNA levels of CD4+ cells in combined therapy vs. 4Mu or anti PD-1; *** *p* < 0.001 mRNA levels of CD8+ cells in combined therapy vs. 4Mu or ** *p* < 0.01 vs. anti-PD-1, one-way ANOVA with Tukey’s multiple-comparison test.

**Table 1 ijms-26-10427-t001:** Fractional tumor volume (FTV) ^1^ relative to untreated controls.

Day ^2^	4Mu (200 mg/kg)	antiPD-1 (100 µg/mL)	Expected ^3^	Observed	R ^4^
6	0.812	0.676	0.832	0.381	2.1
10	0.895	0.423	0.473	0.277	1.7
12	1.331	0.497	0.373	0.304	1.3

^1^ FTV (experimental mean tumor volume)/(control mean tumor volume); ^2^ day after treatment onset; ^3^ (4Mu mean FTV) × (anti PD-1 mean FTV); ^4^ R = [Expected FTV/Observed FTV]. A ratio > 1 indicates a synergistic effect, and a ratio < 1 indicates a less than additive effect.

**Table 2 ijms-26-10427-t002:** The specific primers.

Gene	Primer Reverse (5′‒3′)	Primer Forward (5′‒3′)
mACTb	CACTGTCGAGTCGCGTCC	CCTTCTGACCCATTCCCACC
hACTb	TTTGCAGCTCCTTCGTTGCC	CGCAGCGATATCGTCATCCA
mARG1	CAGAAGAATGGAAGAGTCAG	CAGATATGCAGGGAGTCACC
mhCAIX	GTGGGGACCTCGTGATTCTC	GCAGGGAAGGAAGCCTCAAT
mCD163	GGGATGAGTCCCATCTTTCACTAT	CGAATATCTATGTATCGTGAGCAGACTA
mhCD206	TGTGGTGAGCTGAAAGGTGAC	GTCTTTGTAAATAACCCACCCATCT
mCD4	AGGTGATGGGACCTACCTCTC	GGGGCCACCACTTGAACTAC
mCD8	CCGTTGACCCGCTTTCTGT	CGGCGTCCATTTTCTTTGGAA
mGLUT1	CAATGGCGGCGGTCCTATAA	AACGGACGCGCTGTAACTAT
mIL1b	TGACAGTGATGAGAATGACCTGTTC	TTGGAAGCAGCCCTTCATCT
mIL10	GGTTGCCAAGCCTTATCGGA	ACCTGCTCCACTGCCTTGCT
miNOS	AAGATGGCCTGGAGGAATGC	TGCTGTGCTACAGTTCCGAG
mMCT4	CAAGGCGGACAGAGGCAGATA	CTCTCTCCACAAATGGTGTGC
mPDL1	CCAGCCACTTCTGAGCATGA	AAACATCATTCGCTGTGGCG
mTGFb	ACC AACTATTGCTTCAGCTC	TGTTGGTTGTAGAGGGCAAG
mTNFa	GACCCTCACACTCAGATCATCTTCT	CCACTTGGTGGTTTGCTACGA

## Data Availability

The raw data supporting the conclusions of this article will be made available by the authors on request.

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
