# Peer review of "4-Methylumbelliferone Modulates CAIX to Mitigate Hypoxia-Driven Dysregulation and Enhance PD-1 Immunotherapy in Lung Cancer"

_ijms, 2025, doi:10.3390/ijms262110427_

Round 1

Reviewer 1 Report

Comments and Suggestions for Authors

The manuscript "Modulation of CAIX with 4-methylumbelliferone Mitigates 2 Hypoxia-Induced Metabolic and Immune Dysregulation and 3 synergizes with PD-1 blockade in lung cancer" investigates how 4-methylumbelliferone modulates carbonic anhydrase IX (CAIX) to counteract hypoxia-induced metabolic and immune dysfunction in lung cancer, demonstrating that this intervention can improve the tumor microenvironment. The study further shows that 4-methylumbelliferone synergizes with PD-1 blockade immunotherapy in preclinical models, suggesting its potential as a metabolic adjuvant to enhance cancer immunotherapy efficacy. The following comments can improve the manuscript.

  1. the current title is a bit long and authors may think about one of these "4-Methylumbelliferone Targets CAIX to Improve Hypoxia-Induced Immune Dysregulation and PD-1 Therapy in Lung Cancer" or "4-Methylumbelliferone Modulates CAIX to Mitigate Hypoxia-Driven Dysregulation and Enhance PD-1 Immunotherapy in Lung Cancer"
  2. The abstract lacks mention of any potential side effects or toxicity concerns of 4Mu, which is relevant for therapeutic considerations.
  3. In the introduction, the challenges or limitations of current CAIX inhibitors is missing
  4. Why 4Mu might offer advantages.
  5. Also authors should state the gaps or unresolved questions leading to this study.
  6. Some methods such as small animal numbers (n=5-8) for in vivo experiments may reduce statistical power.
  7. The in vivo anti-tumor efficacy study shows combination synergy but single-agent 4Mu lacks efficacy, raising questions about its standalone therapeutic potential.
  8. Data on potential off-target effects or toxicity of 4Mu in vivo are limited or missing.
  9. The discussion lacks limitations in the clinical cohort and preclinical models.
  10. The translation from murine models to human lung cancer immunotherapy requires cautious interpretation; this caveat could be emphasized more.
  11. There is limited discussion on the possible impact of heterogeneity within NSCLC subtypes on treatment efficacy.

Author Response

Reviewer 1

  1. The current title is a bit long, and authors may think about one of these "4-Methylumbelliferone Targets CAIX to Improve Hypoxia-Induced Immune Dysregulation and PD-1 Therapy in Lung Cancer" or "4-Methylumbelliferone Modulates CAIX to Mitigate Hypoxia-Driven Dysregulation and Enhance PD-1 Immunotherapy in Lung Cancer"

Thank you for your suggestion on improving the title. We have selected the second option and changed the title. Please, find this in the revised manuscript.

  1. The abstract lacks mention of any potential side effects or toxicity concerns of 4Mu, which is relevant for therapeutic considerations.

We thank the Reviewer for raising this important point. We have addressed this issue in the abstract section and added “4Mu was safe and well tolerated…” line 52-53.

  1. In the introduction, the challenges or limitations of current CAIX inhibitors are missing

We thank the Reviewer for this observation. This work was not only to evaluate 4Mu as a CAIX inhibitor (which had been previously reported; doi: 10.1021/jm200983e. Epub 2011 Nov 22) or intended to present it as a better inhibitor than existing options, but also to take advantage of its capacity to potentiate tumor microenvironment modulation. For this reason, the challenges and limitations of other CAIX inhibitors were not mentioned in the Introduction section. However, these aspects related to other coumarins, which also are CAIX inhibitors, are mentioned in the Results section (lines 232-236). Other CAIX inhibitors are mentioned in lines 256-259.

Additionally, we have added a sentence regarding the lack of approval for clinical use of CAIX inhibitors to highlight their limitations (lines 409-410). Furthermore, Supplementary Figure 1D shows the results of acetazolamide, a known CAIX inhibitor, to compare the 4Mu effect on CAIX inhibition in our model.

  1. Why 4Mu might offer advantages.

We thank the Reviewer for this comment. Throughout the manuscript, we have highlighted the advantages of 4Mu treatment, which are also summarized in the Discussion section. Briefly:

  • 4Mu downregulates CAIX, reduces lactate, and restores the extracellular pH, reversing the acidosis generated by hypoxia in the TME.
  • 4Mu polarizes macrophages from the M2-like to the antitumoral M1-like phenotype.
  • 4Mu rescues T cell proliferation suppressed by hypoxia.
  • 4Mu enhances infiltration of T cells in vivo.
  • As previously reported, 4Mu has no antitumor effect when used as However, it shows a synergistic antitumor effect in combination with anti-PD-1 and improves mice survival.
  • 4Mu binds to a different region of the carbonic anhydrase active site, which eventually could lead to testing the combination with sulfonamide inhibitors.

Additionally, in line with the previous comment of the Reviewer, we have mentioned the limitations for clinical use of CAIX inhibitors to highlight the advantage of 4Mu (lines 409-410). More importantly, and in line with the next point  #8 of the Reviewer´s comment, we indicate that 4Mu has a known safety profile and its use is approved in Europe and Asia for hepatobiliary disorders, and it's being tested for other pathologies (lines 381-383).

  1. Also authors should state the gaps or unresolved questions leading to this study.

We thank the Reviewer for this suggestion. We included our aim in the Introduction section (lines 97-99).

  1. Some methods, such as small animal numbers (n=5-8) for in vivo experiments, may reduce statistical power.

We thank the Reviewer for this comment. We used 5–8 mice/group based on previously published data (doi: 10.1038/s41598-021-85491-0, doi.org/10.1158/0008-5472) and on design choices that maximize statistical power and reduce variability while abiding by the 3Rs. Moreover, the observed differences were consistent across two independent experiments (line 563) and reached statistical significance.

  1. The in vivo anti-tumor efficacy study shows combination synergy, but single-agent 4Mu lacks efficacy, raising questions about its standalone therapeutic potential.

We appreciate the Reviewer´s comment. As others and we have previously reported (https://doi.org/10.3389/fonc.2021.710061, https://doi.org/10.1038/mt.2015.112, doi: 10.1038/s41598-024-66914-0), 4-Mu lacks standalone antitumor effect. This was mentioned in the Discussion section, line 387. Multiple studies report enhanced outcomes when 4Mu is combined with other treatments like chemotherapy, adoptive T-cell therapy, or immunotherapy, emphasizing its role as an adjuvant rather than a direct cytotoxic agent. In order to clarify this issue, we have changed the writing of the Results section in line 335-336.

  1. Data on potential off-target effects or toxicity of 4Mu in vivo are limited or missing.

We thank the Reviewer for this comment. In this study, mice were carefully monitored throughout the treatment period for weight loss, behavioral changes, or other signs of distress. No toxicity was observed at the doses used in any experiment. We have introduced a sentence mentioning 4Mu safety in the Results section, lines 340-341.

Additionally, these findings are consistent with our previous preclinical studies in three different mouse strains (BalB/c, C3H, and C57BL/6), which showed no toxicity of 4Mu at the dose used for our experimental models (https://doi.org/10.1038/mt.2015.112;https://doi.org/10.1016/j.ymthe.2018.09.012, https://doi.org/10.1038/s41598-024-66914-0).

More importantly, 4Mu, also known as hymecromone, was approved by the European Medicines Agency (EMA) (https://doi.org/10.3389/fimmu.2015.00123) and has been used in humans for decades as a choleretic with doses up to 3.6 g/day and minimal adverse effects are reported (https://doi.org/10.1172/JCI157983). In this study, the mice were treated with 200 mg/kg, which represents 16.2 mg/kg in human equivalent dose (https://doi.org/10.4103/0976-0105.177703), making the mouse treatment 74 times less than the human dosage. We have introduced a sentence to remark that 4Mu has a known safety profile and its use in humans is approved in Europe and Asia (lines 381-383).

  1. The discussion lacks limitations in the clinical cohort and preclinical models.

We thank the Reviewer for this observation. We addressed the limitations of this study, underscoring its exploratory characteristics in lines 434-438. We acknowledge that patient data could provide further validation, and we are now recruiting patients for a prospective cohort study. Regarding the in vivo experiments, the focus of this work was to test 4Mu in combination with ICIs in controlled preclinical settings, using a model that is reported as a non-responder to immunotherapy. Then, we consider that it is robust to support our conclusions. We have introduced a sentence to highlight this point in the Discussion Section (lines 421).

  1. The translation from murine models to human lung cancer immunotherapy requires cautious interpretation; this caveat could be emphasized more.

We thank the Reviewer´s comment. We are aware of the limitations regarding translation to humans from murine models. While our study focuses on preclinical evaluation of 4Mu’s combinatorial potential, the results provide proof-of-concept evidence that warrants further investigation. We have now emphasized this caveat in the Discussion section to clarify the scope and limitations of our findings (lines 431).

  1. There is limited discussion on the possible impact of heterogeneity within NSCLC subtypes on treatment efficacy.

We note that NSCLC comprises heterogeneous subtypes, primarily LUAD (~50% of NSCLC patients) and LUSC (25-30% of NSCLC patients), which differ in incidence and biology. However, for advanced patients, the standard of care for both subtypes is ICI treatment alone or in combination with chemotherapy, depending on PD-L1 expression in each patient. The LLC model recapitulates the most common lung cancer subtype, LUAD, providing a relevant preclinical platform to evaluate 4Mu in combination with ICI therapy, even though subtype-specific responses may exist. We consider that it would be important to incorporate the assessment of the 4Mu effect in cell lines that represent the LUSC subtype, such as KLN205, in the future. We have addressed this in lines 417-418.

Reviewer 2 Report

Comments and Suggestions for Authors

This manuscript investigates hypoxia-associated metabolic reprogramming (with emphasis on CAIX) in NSCLC and evaluates whether 4-methylumbelliferone (4Mu) can normalize acidosis/lactate, reprogram the tumor microenvironment (TME), and synergize with anti-PD-1 in an LLC model. The topic is clinically relevant. However, there are numerous reporting inconsistencies, figure/legend mismatches, missing methods/statistics details, and flow cytometry presentation issues that currently weaken scientific rigor and reproducibility. Substantial revision is required before the study’s conclusions can be fully supported.

Major points

  1. There are multiple internal inconsistencies in the descriptions of experimental design, figures and figure legends.
    1. For the design of experiment in Fig. 6, in the Results Line 301-302, anti-PD1 is said to be given “3 doses, once a week”. But in Figure 6, anti-PD-1 is shown to be given on day 10, ~day 13 and ~day 17. In addition, in Line 302, the number of mice is said to be “n=8”, but in Methods, the mice number is said to be “n = 5-6/group”.
    2. In Fig. 6A the control curve suggests some mice persisting to ~day 21, while Fig. 6B implies all control animals died by ~day 15.

Please re-check raw data and resolve these inconsistency and revise descriptions accordingly (or provide corrected plots).

  1. The CAIX IHC time points in Fig. 5 are confusing. The text says mice were treated for 8 days, yet “after 24 and 48 h, animals were sacrificed” for CAIX readouts; clarify whether these are 24/48 h after starting 4Mu or after the full 8-day course, and present group-wise quantitative data for all time points/conditions referenced (including control dynamics over time). In addition, how are the CAIX expression quantified in these images?
  2. The flow data in Fig 6 C-D have multiple issues:
    1. The resolution of the representative flow plots in Figure 6 C is very low.
    2. It also shows potential issues with the gating of CD4 and CD8 T cell populations. The T cell gating is clearly contaminated by with events appear to be CD4 and CD8 double positive, which are likely false positive events. The control group is not shown in the plot.
    3. The frequencies shown in the representative flow plots don't match with the statistics in the histogram. For example, flow plot shows 0.3% CD4 in 4Mu group and 5.4% of CD8 in aPD-1+4Mu group, which doesn’t match with the data in the histogram.
    4. The representative flow plot of the control group is missing.
    5. Figure 6D is described as flow data but why the y-axis shown as "Fold change (2ΔΔCT)", which seems to be qPCR data?

  3. TCGA is RNA-seq; please use HGNC gene symbols, not protein names in the corresponding text in the Result. For example, CAIX should be CA9, PD-L1 should be CD274, etc. The current text mixes protein and gene names and could mislead readers.

  4. Many figures are missing key information such as sample size and statistical test used. Please show individual data points in the bar graphs, report the exact tests used and definitions of “independent experiments”.

  5. The narrative implies 4Mu reduces CA9 expression and “restores pH” as a major mechanism. Please strengthen causality by testing whether CA9 knockdown/overexpression phenocopies/abrogates 4Mu effects.

  6. In Fig. 4, the abbreviations on the plots should be clearly annotated in the figure legend. What are these groups in the figures? They are not explained or discussed in the manuscript.

Minor points

  1. Figure 2 legend does not clearly map panels (A/B vs LUAD/LUSC). Please revise legends to explicitly label panels, cohorts, and statistics; ensure statements in text are supported by the shown data.
  2. There are several typos: in Line 96: “vivo” – “in vivo”; in Line 106: “Figure 1 A y B”.
  3. Which figure is Line 109-115 talking about?
  4. Please annotate the gene categories (hypoxia-related, glycolytic/metabolic, immune-related) in Fig 1 D-E.
  5. In Line 205-206, Figure call outs don't match with the figure legends and are confusing.
  6. Please add citations to the statements in Line 232-234.

Author Response

Reviewer 2

This manuscript investigates hypoxia-associated metabolic reprogramming (with emphasis on CAIX) in NSCLC and evaluates whether 4-methylumbelliferone (4Mu) can normalize acidosis/lactate, reprogram the tumor microenvironment (TME), and synergize with anti-PD-1 in an LLC model. The topic is clinically relevant. However, there are numerous reporting inconsistencies, figure/legend mismatches, missing methods/statistics details, and flow cytometry presentation issues that currently weaken scientific rigor and reproducibility. Substantial revision is required before the study’s conclusions can be fully supported.

Major points

There are multiple internal inconsistencies in the descriptions of experimental design, figures, and figure legends.

  1. For the design of the experiment in Fig. 6, in the Results Line 301-302, anti-PD1 is said to be given “3 doses, once a week”. But in Figure 6, anti-PD-1 is shown to be given on day 10, ~day 13, and ~day 17. In addition, in Line 302, the number of mice is said to be “n=8”, but in Methods, the mice number is said to be “n = 5-6/group”.
  2. In Fig. 6A, the control curve suggests some mice persisting to ~day 21, while Fig. 6B implies all control animals died by ~day 15.

Please re-check raw data and resolve these inconsistency and revise descriptions accordingly (or provide corrected plots).

We thank the Reviewer for this keen observation. Regarding point 1, we have changed the Results section (in revised version, line 332-333), indicating the correct days of treatment administration, and corrected the number of mice/groups in line 334 of the revised manuscript. In relation to point 2, the survival curve of the control group was effectively wrong, which is now corrected in Figure 6B.

The CAIX IHC time points in Fig. 5 are confusing. The text says mice were treated for 8 days, yet “after 24 and 48 h, animals were sacrificed” for CAIX readouts; clarify whether these are 24/48 h after starting 4Mu or after the full 8-day course, and present group-wise quantitative data for all time points/conditions referenced (including control dynamics over time). In addition, how is the CAIX expression quantified in these images?

We thank the Reviewer for the comment. The text in lines 324 and 533 of the revised manuscript has been changed to reflect the 24 or 48h treatment with 4Mu. Quantification of CAIX-positive area was done by densitometry using ImageJ software (National Institutes of Health, Bethesda, MD, USA; NIH). We have introduced a sentence in the figure legend (lines 323) and M&M to clarify this point (lines 536-537). Also, Figure 5 now presents data for all time points.

The flow data in Fig 6 C-D have multiple issues:

The resolution of the representative flow plots in Figure 6 C is very low.

Thank you for this comment. We have pasted the flow plots obtained directly from the Accuri software.

It also shows potential issues with the gating of CD4 and CD8 T cell populations. The T cell gating is clearly contaminated by with events appear to be CD4 and CD8 double positive, which are likely false positive events. The control group is not shown in the plot.

Thank you for this comment. We have corrected the cell gating in order to avoid any contamination and also incorporated the plot for the Control group. Please, find this in the new Figure 6.

The frequencies shown in the representative flow plots don't match the statistics in the histogram. For example, flow plot shows 0.3% CD4 in 4Mu group and 5.4% of CD8 in the aPD-1+4Mu group, which doesn’t match the data in the histogram.

We appreciate the Reviewer for having detected this inconsistency.  We have modified the figure for a correct match between the data in the histogram and the representative flow plots (new Figure 6).

The representative flow plot of the control group is missing.

We appreciate this comment. Now we have added it.

Figure 6D is described as flow data, but why is the y-axis shown as "Fold change (2ΔΔCT)", which seems to be qPCR data?

We thank the Reviewer for this remark. The legend in Figure 6D has been corrected to state that the data shown corresponds to qPCR quantification of CD4 and CD8 T cell markers.

TCGA is RNA-seq; please use HGNC gene symbols, not protein names in the corresponding text in the Result. For example, CAIX should be CA9, PD-L1 should be CD274, etc. The current text mixes protein and gene names and could mislead readers.

We thank the Reviewer for this thorough observation. The HGNC gene symbols were correctly used in the revised version of the manuscript.

Many figures are missing key information, such as sample size and statistical test used. Please show individual data points in the bar graphs, report the exact tests used, and definitions of “independent experiments”.

We thank the reviewer for highlighting the lack of this information in the manuscript. We have added a section on statistical analysis in the Materials and Methods section and have also noted it in each figure legend and throughout the Results section. Please find all the changes in the new manuscript version.

The narrative implies 4Mu reduces CA9 expression and “restores pH” as a major mechanism. Please strengthen causality by testing whether CA9 knockdown/overexpression phenocopies/abrogates 4Mu effects.

We appreciate the reviewer’s valuable suggestion. In this work, we focused on the pharmacological inhibition of CAIX using 4Mu, aiming to counteract the hypoxia-driven hostile tumor microenvironment, restore immune competence, and sensitize LLC tumors to immunotherapy. We agree that knockdown/overexpression experiments would provide the most direct demonstration of whether CAIX mediates the effects of 4Mu. While these are beyond the scope of the present study, we performed complementary analyses that support this connection.

Specifically, we tested the CAIX inhibitor acetazolamide (Az). Under hypoxic conditions, CAIX expression was markedly upregulated (p<0.001), and Az treatment significantly reduced CAIX mRNA levels both in normoxia and hypoxia (p<0.01 and p<0.001; Supplementary Fig. 1D). However, this reduction did not translate into measurable functional changes, as assessed by extracellular pH (Supplementary Fig. 1E), in clear contrast with the pronounced effect observed with 4Mu. This suggests that 4Mu exerts a more robust or mechanistically distinct modulation of CAIX compared with Az.

In addition, our transcriptomic analysis (RNA-seq, data not shown) revealed that 4Mu reverses hypoxia-induced gene expression programs linked to CAIX activity, further reinforcing its role as a mediator of 4Mu effects on the hypoxic tumor microenvironment. Together, these data provide indirect but consistent evidence for a functional impact of 4Mu on CAIX. We fully agree, however, that future gain and loss-of-function studies will be essential to definitively establish causality, and we plan to address this in follow-up work.

In Fig. 4, the abbreviations on the plots should be clearly annotated in the figure legend. What are these groups in the figures? They are not explained or discussed in the manuscript.

We thank the Reviewer for this observation. We have added abbreviations on the figure legend and also in the Results section in order to highlight where they are explained (lines 291-301; 277-278; 284-285).

Minor points

Figure 2 legend does not clearly map panels (A/B vs LUAD/LUSC). Please revise legends to explicitly label panels, cohorts, and statistics; ensure statements in text are supported by the shown data.

We appreciate this comment. Now we have revised the legends to clarify the label panel, cohorts, and statistics (lines 1-177)

There are several typos: in Line 96: “vivo” – “in vivo”; in Line 106: “Figure 1 A y B”.

We have corrected these typos (lines 108 and 119).

Which figure is Line 109-115 talking about?

We thank the Reviewer for this observation. We have clarified this issue in the new version of the manuscript (lines 119-122; 131-132)

Please annotate the gene categories (hypoxia-related, glycolytic/metabolic, immune-related) in Fig. 1 D-E.

We thank the reviewer for this suggestion. We have incorporated the gene categories in the new Figure 1 D-E and line 153.

In Line 205-206, Figure call outs don't match with the figure legends and are confusing.

Thank you for the comment. We have clarified this point (lines 226; 228).

Please add citations to the statements in Lines 232-234.

Thank you for this observation. We have incorporated references according to the text, now in line 258.

Round 2

Reviewer 1 Report

Comments and Suggestions for Authors

The authors have responded properly to the comments and the manuscript has been greatly improved and now could be published

Author Response

We are very grateful to the reviewer for recommending our article for publication.

Reviewer 2 Report

Comments and Suggestions for Authors

Thank you for your rapid response. The revised manuscript addresses some of the original points but several core datasets and figure panels have been altered between versions without a clear, verifiable explanation or provision of raw data.

Before the study can be evaluated further, please provide:

  • A full audit trail of every corrected figure/panel, including original raw data files (tumour volume logs, survival records with animal IDs, FCS files with gating, raw qPCR Ct tables, CAIX images with ROI and quantification scripts), file timestamps and the exact steps used to generate the revised figures.
  • Complete gating strategies for all flow cytometry analyses (step‑by‑step plots for each gate), not just final plots.
  • Clear reconciliation of all discrepancies between the text, figure legends, and methods (treatment schedules, time points, group sizes, assay types).
  • Any “RNA‑seq” or other new analyses invoked in the rebuttal made fully available.

Without these materials the reliability of the results cannot be assessed. We therefore cannot evaluate the conclusions until a transparent data package is provided.

Author Response

We have attached the reply for the Reviewer 2 as a pdf file

Round 3

Reviewer 2 Report

Comments and Suggestions for Authors

Thank you for providing additional explanations regarding the discrepancies noted in the previous version. You mention that raw data were supplied as “non-published zip files.” However, I have not been able to locate or access these files within the reviewer system. Without access to the underlying raw data, the information provided in the rebuttal is not sufficient to verify the revised results or resolve key discrepancies between manuscript versions. As such, I am currently unable to provide a further assessment of the manuscript.

Round 4

Reviewer 2 Report

Comments and Suggestions for Authors

Thank you for providing the raw data files and additional explanations. Overall, the newly provided raw data support the the main findings. The overall clarity and organization of the revised manuscript have also improved.